# High Value Utilization of Waste Wood toward Porous and Lightweight Carbon Monolith with EMI Shielding, Heat Insulation and Mechanical Properties

**DOI:** 10.3390/molecules28062482

**Published:** 2023-03-08

**Authors:** Xiaofan Ma, Xiaoshuai Han, Jiapeng Hu, Weisen Yang, Jingquan Han, Zhichao Lou, Chunmei Zhang, Shaohua Jiang

**Affiliations:** 1Jiangsu Co-Innovation Center of Efficient Processing and Utilization of Forest Resources, International Innovation Center for Forest Chemicals and Materials, College of Materials Science and Engineering, Nanjing Forestry University, Nanjing 210037, China; 2Key Laboratory of Green Chemical Technology of Fujian Province University, College of Ecological and Resources Engineering, Wuyi University, Wuyishan 354300, China; 3Institute of Materials Science and Devices, School of Materials Science and Engineering, Suzhou University of Science and Technology, Suzhou 215009, China

**Keywords:** wood, carbon, EMI shielding, heat insulation, mechanical property

## Abstract

With the increasing pollution of electromagnetic (EM) radiation, it is necessary to develop low-cost, renewable electromagnetic interference (EMI) shielding materials. Herein, wood-derived carbon (WC) materials for EMI shielding are prepared by one-step carbonization of renewable wood. With the increase in carbonization temperature, the conductivity and EMI performance of WC increase gradually. At the same carbonization temperature, the denser WC has better conductivity and higher EMI performance. In addition, due to the layered superimposed conductive channel structure, the WC in the vertical-section shows better EMI shielding performance than that in the cross-section. After excluding the influence of thickness and density, the specific EMI shielding effectiveness (SSE/t) value can be calculated to further optimize tree species. We further discuss the mechanism of the influence of the microstructure of WC on its EMI shielding properties. In addition, the lightweight WC EMI material also has good hydrophobicity and heat insulation properties, as well as good mechanical properties.

## 1. Introduction

With the development of technology and the use of electronic devices, electromagnetic (EM) radiation is also increasing. The signal radiation caused by electromagnetic interference (EMI) will interfere with the electronic system [1,2]. This EM radiation will not only cause equipment failure, but also affect human health. The only solution to prevent the harmful EM radiation from damaging electronic equipment is to provide a shield to filter interference. Therefore, in recent years, the research of EMI shielding materials has gradually attracted extensive attention [3]. Among them, the problem of EM radiation in buildings urgently needs extensive attention. In wooden or thin-walled houses, wireless radiation is easier to penetrate. Therefore, the development of materials with EMI shielding performance for buildings and furniture will have extensive application value.

Recently, metals and their oxides [4,5], conductive polymers [6,7] and carbon-based materials [8,9] have made great progress in the field of EMI shielding materials. As an excellent conductor and high thermal conductivity, metal has become a widely used electromagnetic shielding material. However, due to the high density, easy corrosion and difficult processing of metal, its application in lightweight products is limited. Compared with metal materials, polymer-based materials have great advantages in light weight and chemical stability [10,11,12,13]. However, polymer-based materials have the disadvantage of high cost and cannot be recycled [14,15,16,17]. Carbon materials have attracted wide attention in the field of electromagnetic shielding due to their strong conductivity, structural diversity, high stability, light weight and other characteristics [18,19,20]. Graphene, carbon fiber, carbon nanotubes, carbon foam and other new carbon materials have been used in EMI shielding research, but these new carbon materials still have the problem of high cost [21,22,23,24]. Therefore, the development of natural and renewable carbon materials will solve the above problems. In addition, with the aggravation of environmental pollution and the emergence of resource shortage, renewable resources are expected to gradually replace non-renewable resources.

The construction of the city and the demolition of the old building site will produce architectural wooden components. In addition, wood furniture products and other industries will also produce a lot of waste wood. How to deal with and utilize waste wood? It will be very valuable if waste wood is recycled and reprocessed into usable materials. In recent years, wood, as a renewable biomass resource, has gradually realized further functional applications beyond traditional buildings and houses. Wood has the advantages of light weight, low cost, environmental friendliness and easy processing. It is the main carbon source with good graded porous structure [25,26]. The wood is carbonized at a high temperature to form WC material with good conductivity. WC material still maintains the unique hierarchical porous structure of wood [27]. Using wood as the base material for the preparation of EM shielding materials will be beneficial to the functional and high value applications of wood. More importantly, wood-derived carbon (WC) can directly build EMI shielding materials with integrated structure and function without complicated processing technology. At present, some WC composites materials for EMI shielding have been reported [28,29,30,31,32]. These WC composites for EMI shielding mainly use WC as a frame to load highly conductive materials or add magnetic materials. Nevertheless, the EMI shielding mechanism of pure WC has not been analyzed and reported. In addition, the lightweight of EMI shielding materials is very important in the application fields of aerospace, portable and wearable electronic devices. Obviously, the material density has a great impact on the shielding performance of EMI shielding materials [29,33]. Therefore, it is necessary to investigate the EM shielding properties of wood with different densities.

Herein, in order to analyze the EMI shielding mechanism of WC materials, we selected three kinds of wood with different densities (balsa wood, basswood and beech), and prepared WC materials by simple one-step carbonization (Figure 1a). Firstly, the influence of carbonization temperature on the EMI shielding performance of WC materials was investigated. We further explored the influence of different wood sections (cross section and vertical section) and sample thickness on EMI shielding performance. Based on the analysis of the above data, combined with the special hierarchical porous structure of wood, the EMI shielding loss mechanism of wood-based carbon materials is analyzed in detail. Obviously, the inherent hierarchical porous structure of wood is beneficial for the multiple reflection and dissipation of EM waves. In addition, we further tested the hydrophobicity, thermal insulation and mechanical properties of WC materials (Figure 1).

## 2. Results and Discussion

In order to explore the influence of the structure of wood-based derived carbon on EMI shielding properties, the structures of three WC materials were photographed by scanning electron microscopy (SEM). As shown in Figure 2a–c, the cross-section diagrams of wood-derived carbon materials clearly observe the difference in pores in different kinds of wood. The vessel size of balsa wood is larger than the other two kinds of wood, and the porosity is the highest (Figure 2a). The pore structure of basswood is relatively uniform (Figure 2b). Beech is more compact than the other two kinds of wood (Figure 2c). It can be observed from SEM images of cross-section that the WC materials has natural connected channels and regular layered frame structure. This structure has good connectivity, which is very conducive to the conduction of electrons, so that it has high conductivity. When the EM wave passes through this channel structure, it is very beneficial to generate strong EM multiple scattering and reflection [34,35,36,37]. Good vertical channel structure can be seen from the vertical-section of three kinds of WC (Figure 2d–f). We further magnified and observed the surface of the inner wall of the channel of WC material (Figure 2d’–f’). After carbonization, the wood still retains the unique cell wall structure of different kinds of wood. The inner wall of the interconnected channel can be used as an effective EM wave reflecting surface to make the EM wave enter the framework of the carbonized wood and interact with the electrons, resulting in ohmic current loss. The EM wave will repeat the previous loss process when passing through each channel, thus realizing the layered loss of EM wave [29].

The XRD spectra of three wood-derived carbon samples are shown in the Figure 3a. The (0 0 2) plane of graphite has a typical wide diffraction peak at about 24° [38]. This diffraction peak was observed in the XRD spectra of WC samples of three kinds of wood at different carbonization temperatures. This indicates that the carbonization process makes the wood graphitized. With the increase in carbonization temperature, the diffraction peak around 24° gradually becomes sharp, indicating that the degree of graphitization increases. In Raman spectra of Figure 3b, the D band (1340 cm^−1^) and G band (1590 cm^−1^) correspond to defective/disordered carbon and graphitic carbon, respectively. D and G bands are observed in all WC samples, which further indicates that the carbonization process makes the wood graphitized. In three WC samples, the *I*_D_/*I*_G_ ratios of WC-600, WC-800 and WC-1000 all gradually decrease, indicating that carbonization facilitated the graphitization of samples [39]. The results are consistent with the analysis of XRD spectra.

The light weight of WC is very important advantage for applying EM shielding [40]. Wood itself is a natural lightweight biomass. The carbonization process is the reaction of heating and decomposing wood under the condition of isolating air. In the process, natural wood will decompose into tar, water and other substances, while releasing a large amount of gas. Therefore, the volume of wood after carbonization will shrink greatly, and the density will also decrease significantly. The densities of balsa wood, basswood and beech before and after carbonization are recorded in the Figure 3c. After carbonization, the densities of basswood and beech decline substantially. The densities of balsa wood, basswood and beech after carbonization at 1000 °C are 200, 230 and 330 mg cm^−3^, respectively. Obviously, the density of WC materials meets the requirements of lightweight electromagnetic interference shielding material [29,33,41].

Conductivity is important for EM shielding materials [42]. The electrical conductivity of wood-derived carbon increases gradually, with the increase in carbonization temperature (Figure 3d) [43,44]. The interconnected structure of the wood facilitates the rapid transmission of electrons within the WC [45]. With the increase in the graphitization degree in WC, their electrical conductivities are further improved [46]. The WC-1000 of balsa wood exhibits a smaller electrical conductivity (10 S cm^−1^). The conductivity of WC-1000 of bass wood increases to 23 S cm^−1^, and that of WC-1000 of beech increased significantly to 39 S cm^−1^. Obviously, WC with higher wood density exhibits higher electrical conductivity. The pore distributions are analyzed by N_2_ adsorption and desorption experiments (Figure 3e). WC-1000 of balsa wood, basswood and beech all are type-S isothermal curve, which belongs to the pore structure inherent in the wood. According to the pore size of the pore distributions and SEM images of the cross section of the WC materials, the WC materials are mainly composed of macropores.

The self-cleaning performance is also an important advantage for the EMI materials [47]. The contact angles between natural wood and WC were tested. (Figure 3f). The contact angle of natural balsa wood is 53°, and the contact angles of WC-600, WC-800 and WC-1000 after carbonization are 73°, 90° and 97° respectively. The contact angle of natural basswood is 63°, and the contact angles of WC-600, WC-800 and WC-1000 after carbonization are 94°, 99° and 101° respectively. The contact angle of natural beech is 74°, and the contact angles of WC-600, WC-800 and WC-1000 after carbonization are 97°, 101° and 107°, respectively. The contact angles of WC of balsa wood, basswood and beech increases with the increase in carbonization temperature. It can be seen from contact angles that wood-derived carbon all show hydrophobicity, which is essential for EMI materials to achieve self-cleaning [48].

Firstly, the balsa wood with the lowest density was selected to test the EM shielding performance of wood-derived carbon at different carbonization temperature. In order to eliminate the influence of sample thickness on EMI shielding performance, WC samples are uniformly ground to a thickness of 2 mm. The EMI shielding performances of wood, WC-600, WC-800 and WC-1000 of balsa wood at a 2 mm thickness at 8.2–12.4 GHz (X-band) in the cross-section are shown in Figure 4a,b. Obviously, the total shielding effectiveness (SE_T_) of natural wood, WC-600, WC-800 and WC-1000 increases with the increase in carbonization temperature (Figure 4a). The average values of total shielding effectiveness (SE_T_) of the wood and WC-600 are only 0.9 and 15.2 dB, respectively (Figure 4b). The average SE_T_ values of WC-800 and WC-1000 are 18.8 and 22.1 dB, respectively. This is because the enhanced conductivity of WC with the increase in carbonization temperature [48]. In addition, these values of reflection efficiency (SE_R_) and absorption efficiency (SE_A_) of WC samples in the cross-section are calculated by the measured S parameters. The SE_A_ values of WC-600, WC-800 and WC-1000 were 10.4, 10.5 and 14.1 dB, respectively. The SE_R_ values of WC-600, WC-800 and WC-1000, respectively, were 7.0, 8.1 and 7.9 dB. These values of SE_A_ are markedly higher than that of SE_R_.

The average transmission coefficient (T), absorption coefficient (A) and reflection coefficient (R) values are further analyzed (Figure 4c) [36]. For WC-600, WC-800 and WC-1000, the R values are all much larger than the A values, indicating that the EMI mode of WC is mainly reflection. Moreover, in order to evaluate the shielding effect of WC materials on EM waves, the EMI shielding efficiencies of samples are further calculated. The total SE_T_ shielding efficiency of WC-1000 of balsa wood reaches 99% (Figure 4d). By comparing the shielding performance of WC samples at different carbonization temperatures of balsa wood, it can be proved that increasing carbonization temperature can improve the shielding performance of WC samples. Therefore, in subsequent experiments, WC samples with carbonization temperature of 1000 °C were selected to further explore the influence of Wood species, sections and thickness on EMI shielding performance.

In order to further analyze the shielding mechanism, we have tested the shielding properties of three kinds of wood (balsa wood, basswood and beech) at different cross sections. The EMI shielding performances of WC-1000 of balsa wood, basswood and beech at different cross sections (cross section and vertical section) are shown in Figure 5a,b. The cross-sectional average SE_T_ values of WC-1000 (balsa wood, basswood and beech) are 22.1, 25.0 and 26.3 dB, respectively (Figure 5c). Similarly, the sectional average SE_T_ values of WC-1000 (balsa wood, basswood and beech) are 25.1, 27.6 and 28.6 dB, respectively, which also exhibited a significantly increasing trend (Figure 5d). Obviously, WC with higher density has better shielding performance. It may be that the wood with high density has a denser structure after carbonization, which is more conducive to the conduction of electrons and shows a higher conductivity [49]. Conductivity is an important factor of EM shielding materials. In addition, the cross-sectional SE_T_ values of WC-1000 (balsa wood, basswood and beech) are markedly less than that of the vertical-sectional SE_T_ values. This difference may be due to the fact that the tubes inside the wood in the cross-section are parallel to the incident direction of EM waves, resulting in a small amount of EM waves may directly penetrate from the inside of the tubes, forming a wave transmission phenomenon. Evidently, the directional superimposed conductive path of WC materials is conducive to multiple reflection and dissipation of EM waves.

The reflection efficiency (SE_R_) and absorption efficiency (SE_A_) values of WC-1000 samples in the cross section and vertical section are shown in Figure 5c,d. In the cross section, the SE_A_ values of WC-1000 samples (balsa wood, basswood and beech) are 14.1, 14.4 and 17.4 dB, respectively. In the vertical section, the SE_A_ values of WC-1000 samples (balsa wood, basswood and beech), respectively, were 14.4, 19.0 and 20.7 dB. These values of SE_A_ are markedly higher than that of SE_R_. With the increase in carbonization temperature, the SE_A_ values of WC samples increase gradually. Based on the scattering parameters, the reflection coefficient (R), absorption coefficient (A) and transmission coefficient (T) values in cross section and vertical section are calculated (Figure 5e,f). Under different sections, the R values of WC-1000 (balsa wood, basswood and beech) are all much larger than the A values (Figure 5e,f), which further indicates that WC materials are mainly reflect EM waves. According to the above data analysis, the higher the wood density, the higher the shielding efficiency of the WC sample in different wood varieties.

The EMI shielding efficiencies of WC-1000 materials (balsa wood, basswood and beech) at cross section and vertical section are further analyzed. The EMI shielding efficiencies of all samples are higher than 99 % at a thickness of 2 mm (Figure 6a). In cross section, the EMI shielding efficiencies of WC-1000 samples (balsa wood, basswood and beech) are 99.4 %, 99.7 % and 99.8 %, respectively. In vertical section, the EMI shielding efficiencies of WC-1000 samples (balsa wood, basswood and beech) are 99.7 %, 99.8 % and 99.9 %, respectively. It can be clearly seen from the shielding efficiencies of the vertical-sectional WC is higher than that of the cross-section. The shielding efficiency of beech WC-1000 in vertical section reaches 99.9 %, indicating that the material can shield almost all EM waves.

The thickness and density both have an important impact on EMI shielding materials. Therefore, specific EMI shielding effectiveness (SSE/t, SE divided by density and thickness) is often used as a standard to analyze the EMI shielding ability of a material [50]. In cross section, the EMI SSE/t values of WC-1000 samples (balsa wood, basswood and beech) are 550.6 dB cm^2^ g^−1^, 557.6 dB cm^2^ g^−1^ and 400.4 dB cm^2^ g^−1^, respectively. In vertical section, the EMI SSE/t values of WC-1000 samples (balsa wood, basswood and beech) are 627.3 dB cm^2^ g^−1^, 626.4 dB cm^2^ g^−1^ and 424.8 dB cm^2^ g^−1^, respectively. Clearly, SSE/t values of WC-1000 of balsa wood and basswood are higher than that of WC-1000 of beech (Figure 6b). Considering that the average SE_T_ value of basswood is higher than that of balsa wood, the shielding performance of wood-derived carbon materials of basswood is better than that of balsa wood and beech.

The thickness of the material will affect the shielding value, and the EMI shielding performances will improve with the increase in the thickness. The total SE_T_ of WC-1000 of basswood/WC-1000 at different thickness (2.0 mm, 2.5 mm, 3.0 mm, 3.5 mm and 4 mm) are further tested (Figure 6c). Obviously, it can be seen from the total SE_T_ that the WC-1000 samples gradually increase with the increase in thickness. As the thickness increases, the SE_T_ value of WC-1000 of basswood exceeds 40 dB at a thickness of 4 mm. The EMI shielding efficiencies of WC-1000 materials at different thickness (2.0 mm, 2.5 mm, 3.0 mm, 3.5 mm and 4 mm) are further analyzed. The shielding efficiency of WC-1000 of basswood is higher than 99.99% at a thickness of 4 mm, which is sufficient to shield most EM waves (Figure 6d).

The EM shielding mechanism of the WC is simulated in Figure 7. The conductive wood–carbon skeleton contains a large number of free electrons, which can generate current induced by EM field and realize conduction loss [41]. EM wave passing through the directionally superimposed layered conductive frame can accelerate the reflection loss of EM wave. In addition, the hierarchical porous structure of WC has a large internal surface area, which is conducive to the EM wave into the interior and microwave attenuation. EM waves cause various types of scattering inside the pore channel of WC. In addition, due to the regular reticulated layered frame structure of WC materials, EM waves will reflect when contacting the conductive channels of each layer. Then, when the EM waves contact the next layer of channel in WC materials, it will repeat the EM loss process of the previous layer, so as to finally achieve the effect of shielding by reducing the EM wave layer by layer.

In practical application, EMI shielding materials should have good thermal insulation and mechanical properties. Therefore, we further tested the thermal insulation and mechanical properties of wood-derived carbon. Flammable cotton is placed on the surface of wood-derived carbon and heated at the bottom with an alcohol flame (Figure 8a). After heating for 120 s, the cotton maintains a good shape, indicating that the wood-derived carbon has good heat insulation performance. The experimental results show that WC materials as EMI shielding material is expected to adapt to the actual work in extreme high temperature environment. The compressive stress–strain curves of wood-derived carbons are showed in Figure 8b. The compressive stresses of beech/WC-1000, basswood/WC-1000 and balsa wood/WC-1000 are 55.2 MPa, 31.8 MPa and 6.9 MPa, respectively. Obviously, with the increase in density, the mechanical properties of wood-derived carbon are gradually enhanced. The wood-derived carbon of basswood and beech both showed excellent mechanical properties. In general, the carbonized wood (WC materials) retains its unique good mechanical properties. Compared with traditional carbon-based porous materials, it has good mechanical properties [51].

## 3. Materials and Methods

### 3.1. Materials

Balsa wood (*O. pyramidale*), Basswood (*Tilia americana*) and Beech (*Zelkova schneideriana*) were cut from cross and longitudinal sections into cuboids measuring 50 mm × 40 mm × 5 mm. The densities of balsa wood, basswood and beech are 200~250 kg cm^−3^, 500~550 kg cm^−3^ and 700~750 kg cm^−3^, respectively. The moisture content of balsa wood, basswood and beech are 10.0–12.0%, 8.0–12.0% and 8.0–10.0%, respectively.

### 3.2. Fabrication of WC-600, WC-800 and WC-1000

The wood-derived carbon (WC) samples of balsa wood basswood and beech are prepared by carbonizing wood for 1 h at the specified temperature (600, 800, 1000 °C) under N_2_ atmosphere and heating rate (5 °C min^−1^). The products were respectively named WC-600, WC-800 and WC-1000.

### 3.3. Characterizations

Scanning electron microscopy (SEM) (Nova Nano 450, FEI, Hillsboro, OR, USA). Raman spectra (DXR532, Themor, Waltham, MA, USA). Brunauer–Emmet–Teller (BET) method (TriStar II 3020 system, Micromeritics, ASAP2460, Atlanta, Georgia USA). X-ray diffractometer (DY-1291, Philips, Eindhoven, Dutch) with a Cu *K*α line. Four-point probe method (Guangzhou Four-Point-Probe Technology, SDY-4, Guangzhou, Guangdong, China).

The EMI shielding effectiveness was measured with the Vector Network Analyzer (Agilent Technologies N5063A, Palo Alto, State of California, USA) (Appendix A). Schematic sketch of this instrument is showed in Appendix A (see Appendix A). Samples were uniformly polished into 22.86 mm × 10.16 mm cuboids to fit the specific waveguide sample holders in X-band frequency (8.2–12.4 GHz), and analyzed by using the method of waveguide with rectangular regions. EMI shielding materials, via reflection, absorption and transmission, shield EM waves. According to the law of conservation of energy, transmission coefficient (*T*), reflection coefficient (*R*) and absorption coefficient (*A*) can be expressed as:R+A+T=1
$$R={\left|S_{11}\right|}^{2}={\left|S_{22}\right|}^{2}$$
$$T={\left|S_{12}\right|}^{2}={\left|S_{21}\right|}^{2}$$
where *S*_11_ and *S*_21_ denote the reflection and transmission coefficient, respectively. The total EMI shielding effectiveness (*SE_T_*) can, thus, be divided into three aspects: the reflection effectiveness (*SE_R_*), the absorption effectiveness (*SE_A_*) and the multiple reflection effectiveness (*SE_MR_*). *SE_MR_* is generally ignored if the *SE_T_* is more than 15 dB. The shielding effectiveness can be rewritten using the following formulas:$$SE_{R}=10\log\;\left(\frac{1}{1-R}\right)=10\log\;(\frac{1}{1-{\left|S_{11}\right|}^{2}})$$
$$SE_{A}=10\log\;\left(\frac{1-R}{T}\right)=10\log\;(\frac{1-{\left|S_{11}\right|}^{2}}{{\left|S_{21}\right|}^{2}})$$
$$SE_{T}=SE_{R}+SE_{A}$$

EMI shielding efficiency (%) refers to the percentage value used to evaluate the ability to block EM waves, which is obtained from the formula:$$Shielding\;efficiency\;\left(\%\right)=100-(\frac{1}{10^{\frac{SE}{10}}})\times 100$$

In order to fairly compare the actual effectiveness of EMI shielding materials, specific shielding effectiveness (*SSE*) and *SSE*/*t* considering thickness and density are generally used, as shown below:$$\frac{SSE}{t}=\frac{EMI\;SE}{density\times thickness}\left(dB\;cm^{2}\;\cdot \;g^{-1}\right)$$

## 4. Conclusions

As a renewable biomass, wood has a good hierarchical porous structure. Conductive wood carbon skeleton can generate EM field-induced current and realize conduction loss. Directionally stacked layered conductive frame can accelerate the reflection loss of EM waves. The layered porous structure of WC is conducive to EM waves entering the interior and microwave attenuation. In addition, due to the regular reticulated layered frame structure of WC materials, the EM waves will repeat the loss process of the previous layer when contacting the conductive channels of each layer. At the same carbonization temperature and the same thickness, high-density wood with longitudinal section has better EMI performance. However, it is necessary to further analyze SSE/t values of WC after removing the influence of thickness and density. In addition, WC derivatives also show the advantages of light weight, hydrophobic, thermal insulation and high mechanical properties. Therefore, WCs as good EMI shielding materials is expected to replace non-renewable high-cost materials. The research of this work could provide a research basis for the further development of wood-based EMI shielding materials.

## Figures and Tables

**Figure 1 molecules-28-02482-f001:**
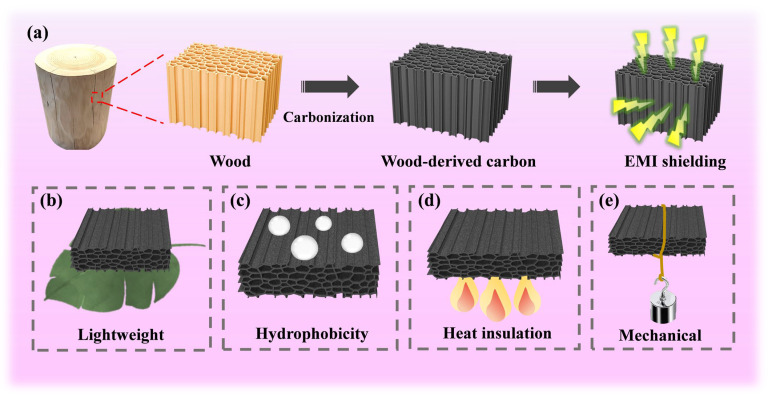
(**a**) Schematic of the preparation of WC for EMI shielding with (**b**) lightweight, (**c**) hydrophobicity, (**d**) heat insulation and (**e**) mechanical properties.

**Figure 2 molecules-28-02482-f002:**
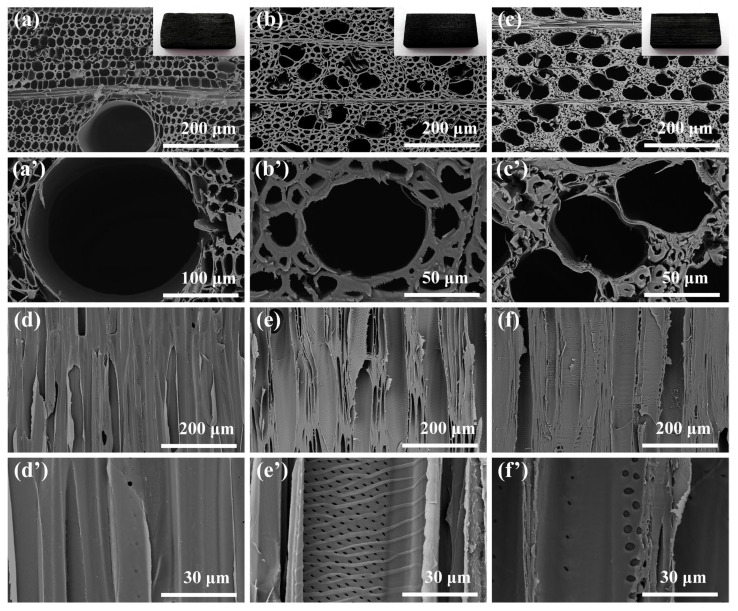
SEM images and pictures of balsa wood-derived carbon (**a**,**a’**), basswood-derived carbon (**b**,**b’**) and beech-derived carbon (**c**,**c’**) in the cross-section, and SEM images of balsa wood-derived carbon (**d**,**d’**), basswood-derived carbon (**e**,**e’**) and beech-derived carbon (**f**,**f’**) in the vertical-section.

**Figure 3 molecules-28-02482-f003:**
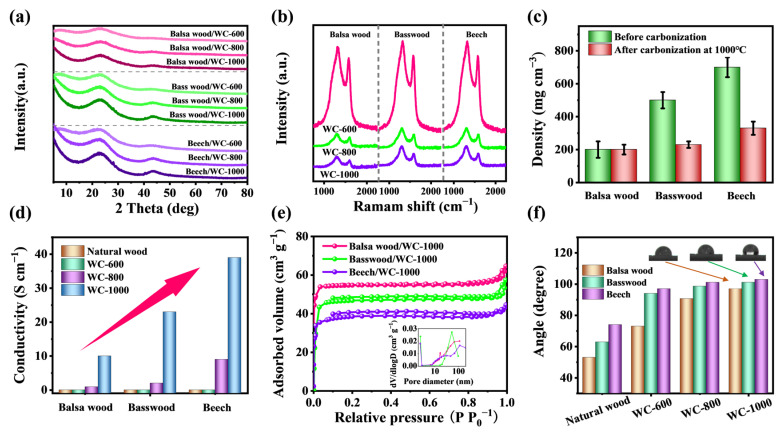
(**a**) XRD spectra and (**b**) Raman spectra of WC-600, CWF-800 and CWF-1000 of balsa wood, basswood and beech. (**c**) Densities and (**d**) electrical conductivities of samples. (**e**) N_2_ adsorption-desorption isotherms and pore size distributions. (**f**) The contact angles of samples.

**Figure 4 molecules-28-02482-f004:**
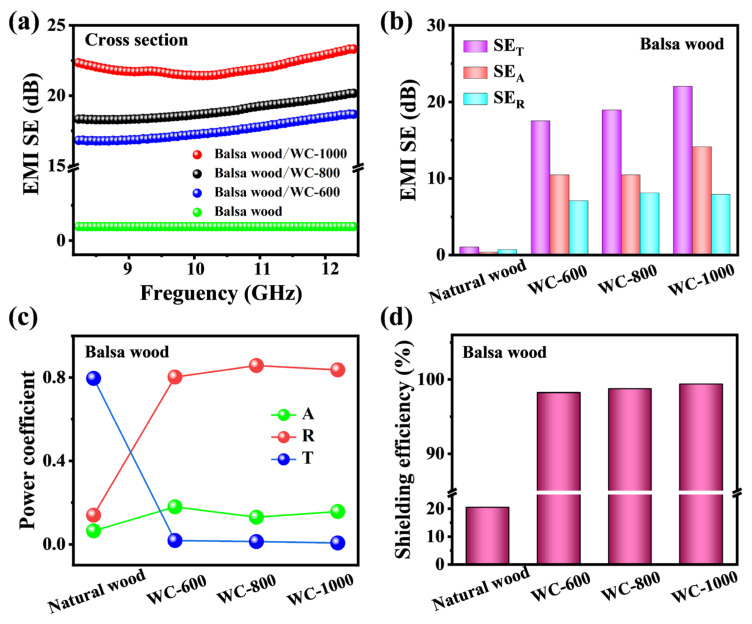
(**a**) EMI SE_T_; (**b**) comparison of the average SE_T_, SE_A_ and SE_R_ values; (**c**) the A, R and T values and (**d**) EMI shielding efficiencies of natural wood, WC-600, WC-800 and WC-1000 of balsa wood.

**Figure 5 molecules-28-02482-f005:**
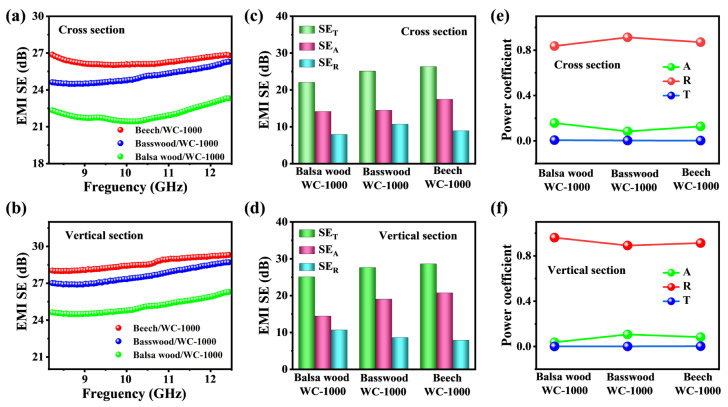
(**a**,**b**) EMI SE_T_; (**c**,**d**) comparison of the average SE_T_, SE_A_ and SE_R_ values; and (**e**,**f**) the A, R and T values of WC-1000 (balsa wood, basswood and beech) under different sections.

**Figure 6 molecules-28-02482-f006:**
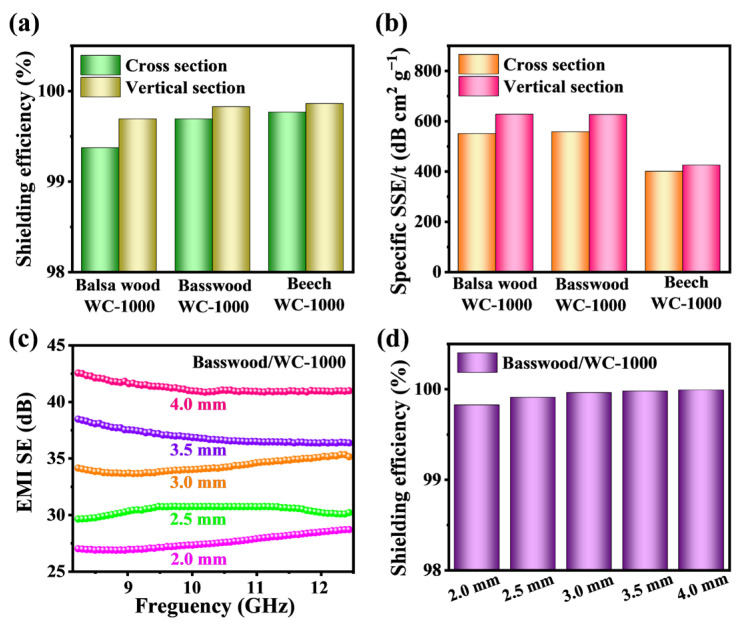
(**a**) EMI shielding efficiencies and (**b**) SSE/t values of WC-1000 (balsa wood, basswood and beech) under different sections. (**c**) EMI SE_T_ and (**d**) EMI shielding efficiencies of WC-1000 of basswood at different thicknesses.

**Figure 7 molecules-28-02482-f007:**
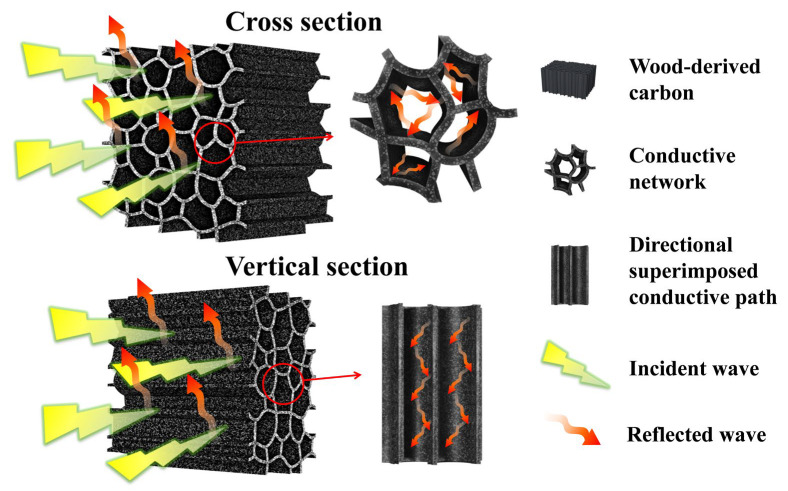
Schematic diagram of EMI shielding mechanism of WC materials.

**Figure 8 molecules-28-02482-f008:**
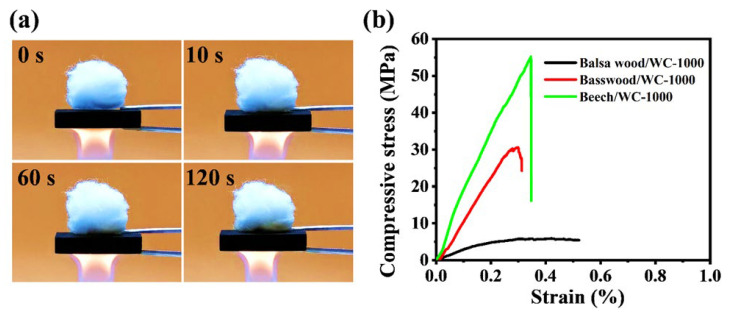
(**a**) Photos of basswood-derived carbon after combustion. (**b**) Compressive stress–strain curves of WC-1000 (balsa wood, basswood and beech).

## Data Availability

The data used to support the findings of this study are available from the corresponding author upon request.

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
