# Peer review of "High Value Utilization of Waste Wood toward Porous and Lightweight Carbon Monolith with EMI Shielding, Heat Insulation and Mechanical Properties"

_molecules, 2023, doi:10.3390/molecules28062482_

Round 1
Reviewer 1 Report
The paper describes the report of conducted experiments in a good way. The research made were well prepared and their results well documented. It is difficult to state about the novelty of the paper as the carbonization of WC is well known and used in various applications.
Particular questions:
1. I was surprised by the layout of the paper – after the relatively short introduction the Results and discussion section comes. It breaks the standard IMRAD layout that should be used in this type of research papers.
2. The paper lacs the pictures/images of the samples that were tested.
3. There is also very little information about the EMI shielding measuring stand in the frequency range of 8.2-12.4 GHz and what was actually measured? No picture, schematic diagram etc.
4. Why was the specific range of tested frequency selected? What is characteristic in the X range that was used for tests? What is the SE value outside the X range?
5. There is also very little information in the Materials and Methods section about other measurement techniques used – no measurement stand schematic/picture and specific conditions required for measurements.
Reviewer 2 Report
Dear Authors,
in my opinion, your manuscript is interesting, quite well prepared, up to date and has a high potential for readers. However, please find below several remarks which should be taken into account before publication:
- regarding the manuscript title: please explain in the Introduction or Materials section, what are unique characteristics of your tested materials (wood) make it - as you said "waste wood"?
- lines 83-90 - I suggest moving (incorporating) it to the Introduction section
- lines 141-142 - when saying that "[...] the density of WC materials meets the requirements of lightweight electromagnetic interference shielding material." - please provide the reference document/standard with the requirements you mentioned
- figure 3(f) - in my opinion, since you tested 3 different wood species, you are not allowed to create the linear plot as 3(f) is, since you did not prove what is between balsa and basswood, as well as between basswood and beech; it is better to make a column plot like 3(d); this remark is valid also to next plots
- figure 4(a) - please provide the X-axis title (Frequency?)
- line 315-319 (Characterizations) - please provide the moisture content of your samples when tested, because, in my opinion, the moisture can have a crucial influence in the case of the features you tested
Best regards!
Round 2
Reviewer 1 Report
Dear Authors,
Thank you very much for the corrections made.
Now, I think that the paper is ready for publication.
And the last thing, please discuss with the editors the need of equations numbering in the final version of the manuscript.